# Factors Associated with Postoperative Complications After Congenital Duodenal Obstruction Surgery: A Retrospective Study

**DOI:** 10.3390/medicina60101722

**Published:** 2024-10-21

**Authors:** Andreea Moga, Radu Bălănescu, Laura Bălănescu, Patricia Cîmpeanu, Mircea Andriescu, Mirela Elena Vasile, Ruxandra Caragata

**Affiliations:** 1Department of Pediatric Surgery, “Grigore Alexandrescu” Clinical Emergency Hospital for Children, 011743 Bucharest, Romania; andreea.moga@ymail.com (A.M.); laura.balanescu@umfcd.ro (L.B.); cimpeanu.patricia@gmail.com (P.C.); mircea.andriescu@umfcd.ro (M.A.); mirela-elena.vasile@rez.umfcd.ro (M.E.V.); ruxandra.caragata@umfcd.ro (R.C.); 2Department of Pediatric Surgery, “Carol Davila” University of Medicine and Pharmacy, 050474 Bucharest, Romania

**Keywords:** duodenal atresia, congenital anomalies, postoperative complications, duodenal obstruction

## Abstract

*Background and Objectives:* Duodenal atresia and stenosis are common causes of intestinal obstruction. Associated anomalies significantly influence early postoperative mortality, while postoperative complications impact long-term survival. *Materials and Methods:* Over a 13-year period from January 2010 to August 2023, a total of 74 infants and children with congenital duodenal obstruction were treated at “Grigore Alexandrescu” Children’s Emergency Hospital and met the inclusion criteria. All patients diagnosed with duodenal obstruction (both instrinsic and extrinsic causes) were included. Analysed data included congenital anomalies, Apgar scores, birth weights, surgical techniques, and complications. *Results:* The associated anomalies included cardiac (*n* = 33), Down syndrome (*n* = 13), neurological (*n* = 11), pulmonary *(n* = 7), renal (*n* = 4), skeletal (*n* = 1), and gastrointestinal and hepatobiliopancreatic anomalies (*n* = 25). In total, 12 patients experienced perioperative ventilation problems. Early postoperative complications (within 30 days) occurred in 21 patients, while 6 had late postoperative complications (after 30 days). Among non-surgical complications, we noted ventilation problems, sepsis (*n* = 7), and pneumothorax (*n* = 1). Surgical complications included adhesive bowel obstruction (*n* = 7), incisional hernia (*n* = 3), peritonitis (*n* = 3), dysfunctional duodenoduodenostomy or duodenojejunostomy (*n* = 3), pneumoperitoneum (*n* = 5), enteric fistula (*n* = 3), and volvulus (*n* = 4). *Conclusions:* Surprisingly, this retrospective study revealed that an Apgar score below 8, along with neurological and pulmonary abnormalities, is associated with postoperative complications. Conversely, other congenital anomalies, low birth weight, and age at admission do not serve as prognostic factors.

## 1. Introduction

Congenital duodenal obstruction (DO) is a common cause of intestinal obstruction in the neonatal period, with an overall incidence of 1 in 5000–10,000 live births [1].

DO is more common in male patients and is often associated with other congenital anomalies, such as Down syndrome, malrotation, and congenital heart disease [2].

Based on the classification proposed by Gray and Skandalakis (Table 1), intrinsic duodenal atresia can be divided into three types, with type I (also known as duodenal web or membrane) being the most common [3]. Duodenal obstruction may also occur due to extrinsic causes such as annular pancreas, preduodenal portal vein, malrotation, or Ladd’s band [4].

Diagnosis is typically made during the prenatal period, although some cases of DO can be detected postnatally within the first hours of life, or later in the case of partial obstruction. Prenatal ultrasound can detect polyhydramnios in up to 81% of cases and reveal a distended stomach and duodenum, characterised by the pathognomonic “double bubble” sign [5,6]. Postnatally, these patients can present with bilious emesis, upper abdominal distension, and aspiration via the nasogastric tube of more than 20 mL of gastric content. Diagnosis is usually confirmed through abdominal ultrasonography, radiography, and upper gastrointestinal contrast studies, which also show the “double bubble” sign [5].

The first successful repair of DO was performed in 1914, at a time when this malformation was considered potentially lethal. Among the causes of death, surgical complications and associated anomalies were the most common. Depending on the type of lesion and associated anomalies, surgical treatment may vary from diamond-shaped duodenoduodenostomy, side-to-side duodenoduodenostomy, duodenojejunostomy to web resection with duodenoplasty [7,8]. The preferred surgical technique has shifted from duodenojejunostomy to side-to-side and then “diamond” duodenoduodenostomy resulting in a significant increase in survival up to 95% [9]. Associated anomalies were found to be the main factor influencing early postoperative mortality, postoperative complications, and long-term survival [10,11,12,13,14,15,16].

This study aims to analyse a cohort of patients diagnosed with duodenal obstruction and to determine whether associated anomalies impact postoperative outcome. We hypothesize that cardiac anomalies and trisomy 21 are the most frequent associated anomalies in patients with DO, and that prematurity and pulmonary anomalies are associated with an increase in postoperative complications.

## 2. Materials and Methods

We conducted a retrospective study and all patients with congenital duodenal obstruction who were treated at “Grigore Alexandrescu” Emergency Hospital for Children over a 13-year period (from January 2010 to August 2023) were included. Data collection was approved by the ethics committee. Inclusion criteria: pediatric patients diagnosed with duodenal obstruction. Exclusion criteria: incomplete data obtained from their chart. Electronical medical records were retrospectively reviewed and data regarding age at admission, sex, gestational, birth weight, prenatal diagnosis, symptomatology, associated anomalies and method of diagnosis, type of duodenal obstruction, surgical approach, early and late postoperative complications, surgical and non-surgical complications and length of hospital stay in days were collected using Microsoft Excel. A total of 77 infants and children were diagnosed with duodenal obstruction, either from intrinsic or extrinsic cause. Of these, 3 patients were excluded from the study due to a lack of essential data, leaving 74 patients for analysis. 

Although duodenal obstruction is a surgical emergency, time should be spent in stabilization and rehydration of the patient. The stomach and proximal duodenum were decompressed using a nasogastric tube and adequate IV fluid resuscitation was initiated. In some cases, a central venous catheter was required. Surgical treatment was determined based on the patient’s anatomical presentation and the surgeon’s clinical judgement.

We hypothesize that associated anomalies are linked with late diagnosis, postoperative complications, and longer hospital stays. Statistical analysis was conducted using the Fisher test and phi coefficient to evaluate associations. A ratio test *p*-value ≤ 0.05 was considered statistically significant.

## 3. Results

From 2010 to 2023, 74 patients (37 boys and 37 girls) with DO were admitted to our department and met the inclusion criteria. None of the patients had a familial history of intestinal malformation. Demographic data are presented in Table 2.

Among the 74 patients, 36 were premature: 26 were born between 35–37 weeks (grade I), 4 between 32–35 weeks (grade II), and 3 before 32 weeks (grade III), with a minimum of 28 weeks and a maximum of 40 weeks. 

Only 18 patients were diagnosed prenatally and polyhydramnios was found in 12 cases. Age at presentation showed a tendency for delayed presentation. Only 9 patients were admitted in the first hours after birth, while 17 patients were admitted in the first and second days of life, and 20 patients were transferred after 7 days of hospitalization in a neonatal ward. The median age at presentation was 4 days, with a maximum of 1890 days.

Birth weight ranged from 950 g to 4250 g with a median of 2570 g. The median Apgar score was 8 (ranging from 5 to 10), with 4 patients having an Apgar score below 7.

Postnatally, patients presented with abdominal distension (*n* = 51), bilious emesis (*n* = 38), hematemesis (*n* = 9) and non-bilious emesis (*n* = 3). Radiological findings included dilated stomach and proximal segment of duodenum on abdominal ultrasonography (*n* = 68) or the “double bubble” sign on abdominal radiography (*n* = 35). In 12 cases, an upper GI contrast study was used to diagnose an incomplete duodenal stenosis. (Figure 1, Figure 2 and Figure 3). As part of the preoperative management, patients underwent further imaging and testing in order to identify associated congenital abnormalities as seen in Table 3. 

Intraoperatively, incomplete obstruction was found in 47 cases as follows: membranous web (*n* = 20), annular pancreas (*n* = 20), Ladd’s band (*n* = 7), while complete obstruction with duodenal atresia was noted in 30 cases. As seen in Table 4, cardiac anomalies were the most common type of associated anomalies encountered in these patients regardless of the type of duodenal obstruction, while gastrointestinal anomales were more frequently found in patients with intrinsec stenosis than in patients with duodenal atresia (Table 4).

Type I duodenal atresia was the most frequently encountered (*n* = 18), while 8 patients were found to have type III and 4 presented with type 4. The defect site was the first segment of the duodenum in 6 cases, the second part of the duodenum in 45 cases, third segment in 21 cases and forth segment in 2 cases. The open approach was favoured, with only 2 cases being operated on laparoscopically.

Data analyses showed that duodenoduodenostomy was the most popular technique (*n* = 34), with concurrent appendectomy performed in only one case, followed by duodenoplasty with web resection (*n* = 17), duodenojejunostomy (*n* = 17), and Ladd’s procedure (*n* = 9). The choice of technique was made after intraoperative diagnosis. Intestinal malrotation was identified as a cause of duodenal obstruction in 6 cases and as an associated anomaly in 9 cases. (Table 5),

In total, 21 patients had early postoperative complications (within the first 30 days) and 6 had late postoperative complications (after 30 days). Among the non-surgical complications, we encountered ventilation problems (*n* = 12), sepsis (*n* = 7), and pneumothorax (*n* = 1). Surgical complications included adhesive bowel obstruction (*n* = 7), incisional hernia (*n* = 3), peritonitis (*n* = 3), dysfunctional duodenoduodenostomy or duodenojejunostomy (*n* = 3), pneumoperitoneum (*n* = 5), enteric fistula (*n* = 3), and volvulus (*n* = 4). 

Due to postoperative complications, 12 patients required a second surgery. Duodenoduodenostomy stenosis was found in 2 cases and was resolved endoscopically. One dysfunctional duodenojejunostomy was resolved with gastrojejunostomy and duodenal exclusion. Overall survival was 100%. 

To determine if patient-related factors impacted postoperative outcomes, we looked at all associated anomalies and calculated Phi coefficient and *p* value for early and late postoperative complications. In selected cases, correlations with sepsis, pneumoperitoneum, and adhesive bowel obstruction were also calculated (these were the most frequent complications). 

The results, as shown in Table 6, highlight a strong association between neurologic and pulmonary anomalies and postoperative complications. Neurologic anomalies are associated with both early (*p* value < 0.01) and late complications (*p* value= 0.011), more precisely with sepsis (*p* < 0.01), pneumoperitoneum (*p* = 0.003), enteral fistula (*p* = 0.01) and adhesive bowel obstruction (*p* < 0.01). Pulmonary anomalies were strongly associated with early postoperative complications (n < 0.01), with sepsis (*p* < 0.05) and anastomotic dysfunction (0 < 0.05). Ventilation problems were recorded and associated with early postoperative complications (*p* < 0.01), sepsis (*p* < 0.05), enteral fistula (*p* < 0.01), and pneumoperitoneum (*p* = 0.005).

We also analysed if the gender, Apgar score or birth weight may impact the postoperative outcome and discovered that male gender is associated with early postoperative complications, an Apgar score lower or equal to 7 is associated with early and late postoperative complications, sepsis, adhesive bowel obstruction and reoperation. Birth weight was not associated with any complications.

## 4. Discussion

Duodenal obstruction is a frequent anomaly diagnosed in the neonatal period and occasionally in infants or small children [10,17]. DO is commonly associated with other anomalies such as cardiac malformations, Down syndrome, and VACTERL syndrome [18,19,20].

Congenital duodenal obstruction is a clinically and embryologically interesting condition, with its aetiology categorised into intrinsic and extrinsic causes. Intrinsic duodenal obstruction is caused by atresia, stenosis, and duodenal diaphragm; extrinsic obstruction is caused by malrotation with Ladd’s bands, annular pancreas or preduodenal vein. It is important to note that these causes may be associated. As seen in the literature, we also found Ladd’s bands associated with intrinsic stenosis, web, or atresia [20]. This association underscores the importance of evaluating intestinal permeability even after finding an extrinsic cause of obstruction. 

We compared our results with those previously published. Patients’ characteristics results were similar to other studies: median birth weight of 2570 g versus 2495 g in a study from Italy and 2530 g in a study from Tokyo [19,21]. Almost half of the study population were preterm babies (48.8% versus 48%) [19].

Our study found a low rate of prenatal diagnosis rate at 24.4%, significantly lower than the 62% reported in other studies [21]. Prenatal screening is a national challenge in Romania [22]. Many factors were included: national funding, availability of services providers, medical education of pregnant women, patient access to prenatal services and doctors’ training in prenatal diagnosis. Although duodenal obstruction screening can be achieved through cost-effective and accessible ultrasound, many women in Romania have uninvestigated pregnancies due to a lack of medical education or financial means. A cross-sectional study conducted in Romania from 2019 to 2020 involving a total of 200 pregnant women proved that teenage mothers face higher hospitalization rates during pregnancy, increased fetal health problems, low level of medical education, and reduced accessibility to medical services during pregnancy [23].

The median age at admission for our patients was 4 days, with a minimum of 0 days and a maximum of 1890 days, i.e., more than 5 years. While this seems surprising, particularly considering the gravity inherent to the topic of congenital malformation, it is not uncommon for diagnosis to be made later than the perinatal period. Incomplete stenoses (Ladd’s band or incomplete diaphragm) may cause mild symptoms or patients may even be asymptomatic. Instances of intestinal malrotation have even been diagnosed in adulthood [24].

Postnatal diagnosis was established using clinical features and imaging results. Bilious versus non-bilious emesis is an important detail that suggests the site of obstruction. 

Radiological investigations traditionally start with a simple abdominal X-ray study; however, ultrasound has become increasingly popular for diagnosing intestinal obstruction has gained popularity due to its dynamic examination capabilities, low cost, and absence of radiation [5,25]. Contrast ultrasound examination may be an interesting option in duodenal incomplete stenosis. Annular pancreas is a frequent cause of duodenal obstruction and the primary diagnostic investigation is an abdominal ultrasound. CT scan that is rarely used for confirmation [26].

The treatment of congenital duodenal obstruction depends on the type of obstruction. The choice of surgical technique has remained a subject of interest over the past 50 years. Duodenojejunostomy, a latero-lateral anastomosis between the proximal duodenum and jejunum, was once a favoured technique due to its simplicity. However, it has become less popular because of potential postoperative complications such as duping syndrome, anastomosis stenosis, blind loop syndrome, and marginal ulcers. Several reports have shown that duodeno-duodenostomy produces the best results with earlier oral feeds and less complications. Although this technique was described in 1977 by Kimura, it is not the preferred choice among surgeons in our clinic [27]. In our cohort, 17 patients underwent duodeno-jejunostomy, with complications arising from ulceration at the level of the anastomosis, necessitating a gastroenteroanastomosis. Duodeno-jejunostomy is considered less physiologically disruptive and this is probably why most surgeons choose to perform a duodeno-duodenostomy. Studies show that gastrojejunal anastomosis is associated with a high incidence of marginal ulceration and bleeding and thus should be avoided [28,29]. No gastrostomy or jejunostomy were performed in our studies because it is not considered useful. Tactical appendectomy was carried out in only one case, so it is not routinely performed in our clinic. Treatment of intrinsic obstruction secondary to intraluminal diaphragm can be either duodenoplasty or diaphragm resection (*n* = 17). 

The Ladd procedure is the standard technique for intestinal malrotation and involves the division of peritoneal bands, reduction of intestinal volvulus, and functional positioning of the intestine (in a non-rotation position). It was performed in 15 cases, of which, it is worth mentioning that in 9 cases the patients also had an intrinsic obstruction so the malrotation was an associated anomaly, not the main cause of obstruction. 

Regarding the duodeno-duodenostomies, strictures were reported, which were resolved by endoscopic dilatation. A study conducted in Shanghai proved that endoscopic balloon-dilatation after membrane resection technique is safe, effective, and feasible for membranous duodenal stenosis [28]. In our clinic, we performed balloon dilatation for secondary stenosis, but never as a first line treatment for duodenal obstruction. 

Lately, a particular interest has been given to associated factors that can significantly influence patient prognosis. Studies have shown that the greatest association involving duodenal atresia is between cardiac malformations and Down syndrome [12,18,19,30]. Of course, in our study these were also found to be the most common associated abnormalities, but we were able to show that they do not impact the patient’s adverse outcome as we would have expected. Fortunately, no deaths were recorded in the study group, but postoperative complications posed a real mortality risk and represent a cause for morbidity.

We calculated whether the surgical technique was associated with any type of complications (of those found in the study group) and found that it is not the technique that matters, but the patient factors, i.e., patient gender, Apgar score at birth and associated anomalies. As presented above, we demonstrated that neurological and pulmonary abnormalities are associated with postoperative complications. In addition, we observed an association between ventilatory disorders, (which is itself a medical complication) and surgical complications, such as pneumoperitoneum and enteral fistula, which may suggest that the mechanism behind these complications may be ischemic and there is a possibility that ventilatory disorders are the cause, not just a risk factor; however, this remains a subject to be studied. The mortality rate (0%) was lower than reported in other studies, but complication rates were high. The most frequent postoperative complications were sepsis, intestinal occlusion, and pneumoperitoneum. Other authors even chose to divide complications into anastomotic and non-anastomotic [31,32]. We must mention that in our study, there were three cases classified as anastomosis dysfunction (two duodeno-duodenostomies and one duodeno-jejunostomy).

## 5. Conclusions

In summary, this study on congenital duodenal obstruction demonstrated that poor outcomes are strongly associated with a low Apgar score at birth, as well as neurologic and pulmonary anomalies. We also found a correlation between ventilation problems and surgical complications. We believe that future studies should focus on the possible mechanisms behind these associations to determine whether there is a direct causal link between them.

## Figures and Tables

**Figure 1 medicina-60-01722-f001:**
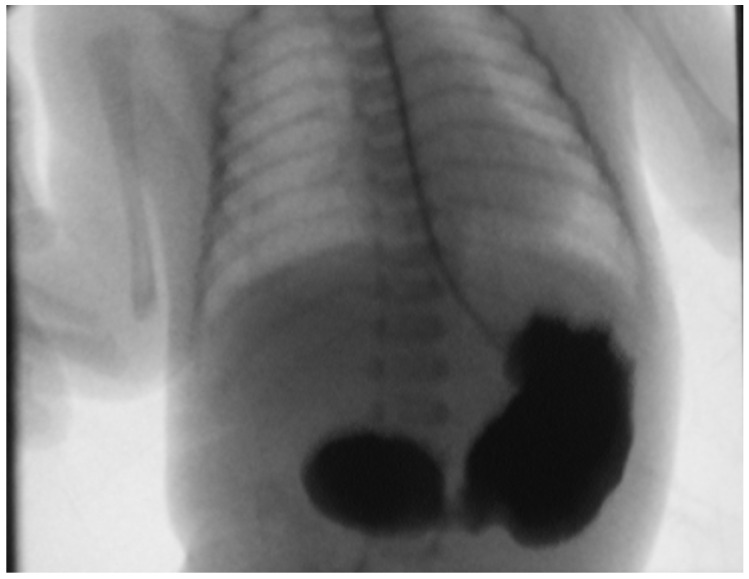
Upper GI contrast study in a case of duodenal atresia showing the characteristic double bubble sign. The contrast material reveals a dilated stomach and duodenum.

**Figure 2 medicina-60-01722-f002:**
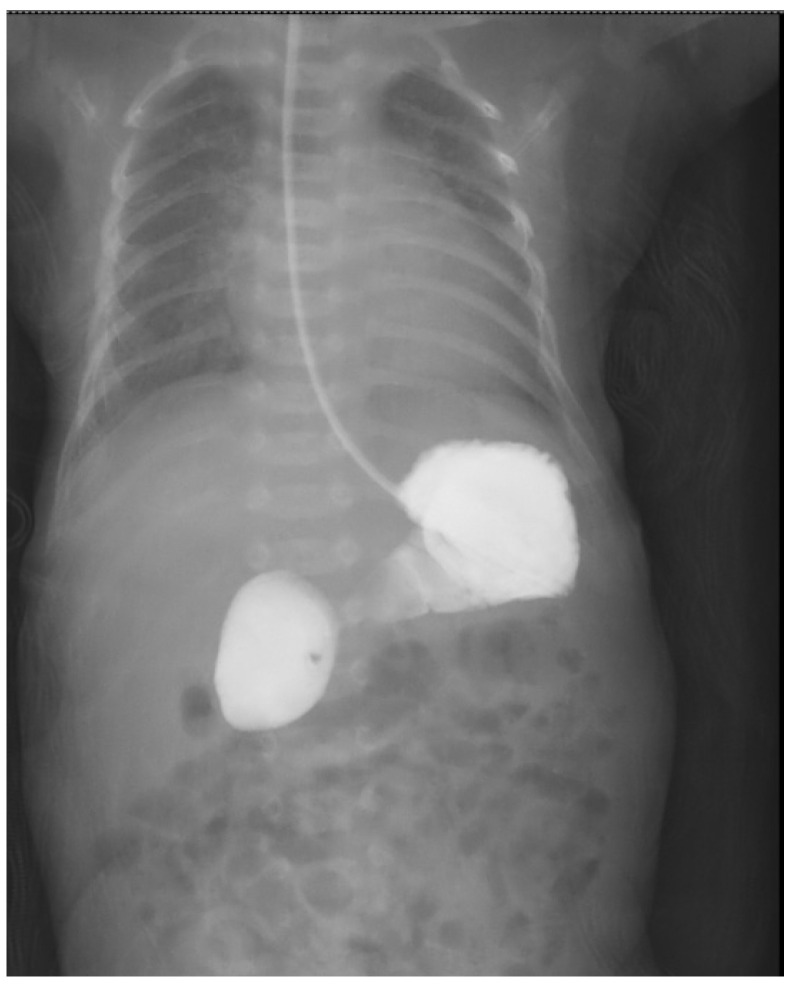
Upper GI contrast study showing incomplete duodenal obstruction (incomplete diaphragm) immediately after administration of 10 mL of contrast material.

**Figure 3 medicina-60-01722-f003:**
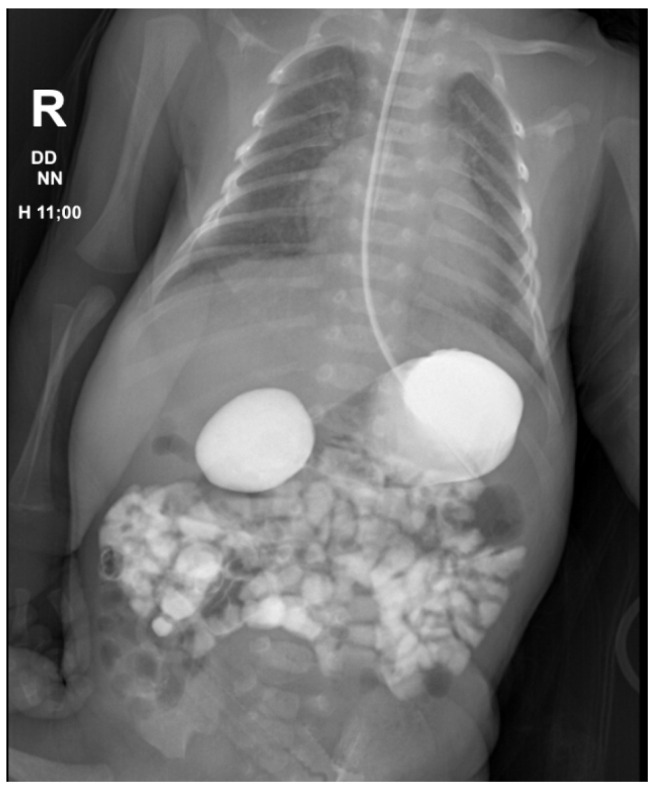
Upper GI contrast study demonstrating incomplete duodenal obstruction (incomplete diaphragm) 2 h after contrast administration.

**Table 1 medicina-60-01722-t001:** Gray and Skandalakis classification of duodenal atresia [3].

Type I (92% of cases)	A web formed by mucosa and submucosa is present with no defect of the muscle layer; a windsock deformity may occur if the web is thin.
Type II (1% of cases)	The duodenal ends are atretic, separated by some distance but attached by a cord; the mesentery is intact.
Type III (7% of cases)	The duodenal ends are atretic, separated by some distance but without any connecting tissue; V-shaped defect in the mesentery.

**Table 2 medicina-60-01722-t002:** Clinical features.

Clinical Features	N = 74	Average	Percentage
GA (weeks) *	range 27 to 40 weeks	36 weeks	
Preterm	36	34.11	48.64%
Early preterm < 34 weeks	8	30.62	
Birth weight *	range 950 g to 4250 g	2570 g	
Birth weight < 2500 g	31	1857.74 g	41.89%
Birth weight < 2000 g	18	1606.11 g	24.32%
Prenatal diagnosis	18		24.32%
Polyhidramnios	12		16.21%
Associated anomalies	50		67.56%

GA: Gestation age. * median value.

**Table 3 medicina-60-01722-t003:** Associated anomalies.

Categories	Anomalies	N	Percentage
Cardiac		33	44.59%
	Atrioventricular septal defect	1	1.35%
	VSD	4	5.40%
	ASD	16	21.62%
	PDA	17	22.97%
	CoA	2	2.70%
	Pulmonary valve stenosis	3	4.05%
	ASA	4	5.405%
Chromosomal	21 trisomy	13	17.56%
Gastrointestinal and hepatobiliopancreatic		25	33.78%
	Intestinal rotational abdormalities	9	12.16%
	Anorectal malformation	1	1.35%
	Hirschsprung’s disease	1	1.35%
	Intestinal duplication	1	1.35%
	Annular pancreas	17	22.97%
	Billiary malformations	3	4.05%
Pulmonary		7	9.45%
	Pulmonary hypertension	4	5.40%
	Pulmonary hypoplasia	3	4.05%
Renal		4	5.40%
	Hydronephrosis	3	4.05%
	Renal ptosis	1	1.35%
Neurologic		11	14.84%
	Mycrocephalia	1	1.35%
Skeletal		1	1.35%
	Cervical	1	1.35%

VSD: Ventricular septal defect; PDA: patent ductus arteriosus; ASD: atrial septal defect; CoA: Coarctation of the aorta; ASA: atrial septal aneurysm.

**Table 4 medicina-60-01722-t004:** Distribution of associated anomalies by cause of duodenal obstruction.

Associated Anomalies	Intrinsic Stenosis	Annular Pancreas	Atresia
Cardiac	7 (35%)	9 (45%)	17 (56.66%)
Neurological	1 (5%)	3 (15%)	4 (13.33%)
Pulmonary	1 (5%)	0	2 (6.66%)
Gastrointestinal and hepatobiliary	5 (25%)	5 (25%)	3 (10%)
Renal	3 (15%)	0	2 (6.66%)
21 trisomy	4 (20%)	5 (25%)	4 (13.33%)

**Table 5 medicina-60-01722-t005:** Surgical treatment.

Operative Procedure	N
Duodeno-duodenostomy	33
Duodeno-duodenostomy and appendecectomy	1
Excision of duodenal diaphragm and duodenoplasty	17
Ladd’s procedure	15
Duodeno-jejunostomy	17

**Table 6 medicina-60-01722-t006:** Association between patient anomalies and postoperative complications.

x	y	Phi Coefficient	*p* Value
Down syndrome	Early postoperative complications	0.0260	0.248
	Late postoperative complications	0.1230	0.289
Cardiac	Early postoperative complications	0.1589	0.171
	Late postoperative complications	0.1319	0.256
Neurologic	Early postoperative complications	0.4953	0.00002
	Late postoperative complications	0.2933	0.011
	Sepsis	0.6437	<0.01
	Pneumoperitoneum	0.3415	0.003
	Enteral fistula	0.2993	0.01
	Anastomosis dysfunction	0.1067	0.358
	Intestinal obstruction	0.5139	<0.01
Pulmonary	Ealy postoperative complications	1	<0.01
	Late postoperative complications	0.0610	0.400
	Sepsis	0.4018	<0.05
	Pneumoperitoneum	0.2176	0.06
	Intestinal obstruction	0.1676	0.149
	Anastomosis dysfunction	0.3051	<0.05
Ventilation problems	Sepsis	0.4841	0.0003
	Pneumoperitoneum	0.3197	0.005
	Enteral fistula	0.4672	<0.01
	Anastomosis dysfunction	0.0954	0.411
Gastrointestinal	Early postoperative complications	0.0945	0.583
	Late postoperative complications	0.1241	0.714
Renal	Early postoperative complications	0.0179	0.122
	Late postoperative complications	0.0710	0.458
Apgar score ≤ 7	Early complications	0.2923	0.011
	Late complications	0.4065	0.000047
	Reoperation	0.3037	0.008
	Sepsis	0.2336	0.044
	Adhesive bowel obstruction	0.3588	0.002
Male gender	Early postoperative complicatons	0	<0.01
	Late postoperative complications	0	1

## Data Availability

The data presented in this study are available on request from the corresponding author due to ethical considerations.

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
