# Peer review of "Factors Associated with Postoperative Complications After Congenital Duodenal Obstruction Surgery: A Retrospective Study"

_medicina, 2024, doi:10.3390/medicina60101722_

Round 1

Reviewer 1 Report

Comments and Suggestions for Authors

This study has many structural and methodological problems, which are mentioned below. The article should be carefully reviewed and rewritten:

1- In the abstract, line 11, delete the word Introduction.

2- The abstract should start with an introduction that deals with the importance of the topic under study and the purpose of the study in a few lines. Therefore, remove statistics and definitions from the beginning of the abstract.

3- If your abstract  is unstructured, remove the headings of testers such as Materials and methods, Results and conclusion from the text label. If structured, separate with:

4-In the summary, line 18, delete the study year range that you mentioned above.

5-Keywords are based on mesh. Also reduce the number of keywords.

6-Page 2, line 71, start Materials and Methods with a short description related to the type of study.

7- Mention the inclusion and exclusion criteria.

8-In Table 1 and Table 2 and Table 3, It is not enough to mention the number, you must express the average and percentage.

9-The discussion is very incomplete and does not address important findings. It should be rewritten.

10- Number 5 and 6 is the conclusion?? Please correct it.

11- Add the limitations section to the article.

Comments on the Quality of English Language

This study has many structural and methodological problems, which are mentioned below. The article should be carefully reviewed and rewritten:

1- In the abstract, line 11, delete the word Introduction.

2- The abstract should start with an introduction that deals with the importance of the topic under study and the purpose of the study in a few lines. Therefore, remove statistics and definitions from the beginning of the abstract.

3- If your abstract  is unstructured, remove the headings of testers such as Materials and methods, Results and conclusion from the text label. If structured, separate with:

4-In the summary, line 18, delete the study year range that you mentioned above.

5-Keywords are based on mesh. Also reduce the number of keywords.

6-Page 2, line 71, start Materials and Methods with a short description related to the type of study.

7- Mention the inclusion and exclusion criteria.

8-In Table 1 and Table 2 and Table 3, It is not enough to mention the number, you must express the average and percentage.

9-The discussion is very incomplete and does not address important findings. It should be rewritten.

10- Number 5 and 6 is the conclusion?? Please correct it.

11- Add the limitations section to the article.

Author Response

Thank you for your review. We adapted the abstract as you recommended and we reduced the number of keywords. Also we reorganised the Materials and Methods in order to  mention the type of the study, the inclusion and exclusion criteria. We modified table 1,2 and 3 and we expressed the average and the percentage.

Reviewer 2 Report

Comments and Suggestions for Authors

The introduction is concise but could benefit from the inclusion of more relevant references, especially concerning previous studies related to postoperative outcomes in pediatric duodenal obstruction surgeries. Kindly provide more detail on the statistical methods used, particularly explaining the selection of the Fisher test and phi coefficient for your analysis.

Author Response

Thank you for your review. We tried to revise the article as per your suggestions. 

Reviewer 3 Report

Comments and Suggestions for Authors

This paper is a retrospective study examining factors related to postoperative complications of duodenal atresia. However, there are several significant concerns with the manuscript:

  1. The statistical analysis is limited to univariate analysis, which is insufficient. Given the retrospective nature of the study, there are many potential biases, and caution is required in interpreting the results. Multivariate analysis or propensity score matching should be utilized to address these biases.

  2. 'Hypotonia' is not a diagnosis but rather a clinical finding. Therefore, it is inappropriate to list it as a associated anomalies.

  3. The need for careful postoperative management of patients with duodenal atresia who have associated malformations, regardless of neurological or pulmonary complications, is not a novel finding. This paper lacks originality in its conclusions.

Author Response

Thank you for your review. We adapted the article as per your suggestions. 

Reviewer 4 Report

Comments and Suggestions for Authors

The manuscript is interesting, however some questions are raised.
Authors should describe the surgical technique better. Did all the patients have a D-D anastomosis? Were all the cases done open?
Numbers do not match in the abstract and in the result section. Total number of patients do not add up to 77. In the gestational age descriptions, not all patients have a gestational age category (totals do not add up). It is not clear how a duodenal obstruction could present at 1890 days, unless it is an annular pancreas that let some of the fluid go through. This patient should probably have been excluded from the analysis. There is no IRB statement of ethics approval protocol number. The discussion is very thorough and could potentially be shortened. Comments on the Quality of English Language

Few spelling and grammatical errors should be corrected. 

Author Response

Thank your for your review. We added details to the surgical techniques and made the changes in the abstract, The total of included patients is 74. The patient that presented at 1890days was  annular pancreas. We included this patient because we considered it a particularity of duodenal incomplete obstruction.

Round 2

Reviewer 1 Report

Comments and Suggestions for Authors

The article was revised. The corrections are well done by the authors and can be published in the journal.

good luck

Comments on the Quality of English Language

The article was revised. The corrections are well done by the authors and can be published in the journal.

good luck

Reviewer 3 Report

Comments and Suggestions for Authors

The authors corrected the manuscript appropriately.